# Health care services use, stillbirth, and neonatal and infant survival following implementation of the Maternal Health Voucher Scheme in Bangladesh: A difference-in-differences analysis of Bangladesh Demographic and Health Survey data, 2000 to 2016

Arijit Nandi[1,2]*, Thomas J. Charters[1], Amm Quamruzzaman[3], Erin C. Strumpf[1,2], Jay S. Kaufman[1,2], Jody Heymann[4], Arnab Mukherji[5], Sam Harper[1,2]

1 Institute of Health and Social Policy, McGill University, Montréal, Québec, Canada, 2 Department of Epidemiology, Biostatistics and Occupational Health, Montréal, Québec, Canada, 3 Interdisciplinary Studies Field, University of California, Berkeley, California, United States of America, 4 WORLD Policy Analysis Center, Fielding School of Public Health, University of California, Los Angeles, California, United States of America, 5 Indian Institute of Management Bangalore; Centre for Public Policy, Bengaluru, India

* arijit.nandi@mcgill.ca

## Abstract

### Background

Starting in 2006 to 2007, the Government of Bangladesh implemented the Maternal Health Voucher Scheme (MHVS). This program provides pregnant women with vouchers that can be exchanged for health services from eligible public and private sector providers. In this study, we examined whether access to the MHVS was associated with maternal health services utilization, stillbirth, and neonatal and infant mortality.

### Methods and findings

We used information on pregnancies and live births between 2000 to 2016 reported by women 15 to 49 years of age surveyed as part of the Bangladesh Demographic and Health Surveys. Our analytic sample included 23,275 pregnancies lasting at least 7 months for analyses of stillbirth and between 15,125 and 21,668 live births for analyses of health services use, neonatal, and infant mortality. With respect to live births occurring prior to the introduction of the MHVS, 31.3%, 14.1%, and 18.0% of women, respectively, reported receiving at least 3 antenatal care visits, delivering in a health institution, and having a skilled birth attendant at delivery. Rates of neonatal and infant mortality during this period were 40 and 63 per 1,000 live births, respectively, and there were 32 stillbirths per 1,000 pregnancies lasting at least 7 months. We applied a difference-in-differences design to estimate the effect of providing subdistrict-level access to the MHVS program, with inverse probability of

**Data Availability Statement:** Analyses utilized two sources of secondary data, specifically: (1) pregnancy and live birth information collected from respondents surveyed as part of the Bangladesh Demographic and Health Surveys (DHS) and (2) upazila-level data on socio-demographic and other indicators from the World Bank's Bangladesh Interactive Poverty Maps 2010. The DHS data are publicly available, but users must first register with the DHS program. Registration, which requires a summary of the proposed study and selection of country datasets, can be completed at: http://www. dhsprogram.com/data/new-user-registration.cfm. The World Bank data are available for download here: http://www.worldbank.org/en/news/feature/ 2014/09/30/poverty-maps The statistical code for producing the analytic dataset and replicating our results are available with unrestricted access from the Harvard Dataverse: https://dataverse.harvard. edu/dataverse/3po.

**Funding:** All authors acknowledge funding from the Canadian Institutes of Health Research Operating Grant, "Examining the impact of social policies on health equity" (ROH-115209). AN was supported by the Canada Research Chairs program. The funders had no role in study design, data collection and analysis, interpretation of data, preparation of the manuscript, or decision to submit the paper for publication.

**Competing interests:** The authors have declared that no competing interests exist.

**Abbreviations:** CI, confidence interval; BDHS, Bangladesh Demographic and Health Survey; DD, difference-in-difference; DHS, Demographic and Health Survey; EA, enumeration area; GDP, gross domestic product; ITT, intention-to-treat; LMIC, low- and middle-income country; MHVS, Maternal Health Voucher Scheme; MOHFW, Ministry of Health and Family Welfare; SDG, Sustainable Development Goal.

treatment weights to address selection into the program. The introduction of the MHVS program was associated with a lagged improvement in the probability of delivering in a health facility, one of the primary targets of the program, although associations with other health services were less evident. After 6 years of access to the MHVS, the probabilities of reporting at least 3 antenatal care visits, delivering in a health facility, and having a skilled birth attendant present increased by 3.0 [95% confidence interval (95% CI) = −4.8, 10.7], 6.5 (95% CI = −0.6, 13.6), and 5.8 (95% CI = −1.8, 13.3) percentage points, respectively. We did not observe evidence consistent with the program improving health outcomes, with probabilities of stillbirth, neonatal mortality, and infant mortality decreasing by 0.7 (95% CI = −1.3, 2.6), 0.8 (95% CI = −1.7, 3.4), and 1.3 (95% CI = −2.5, 5.1) percentage points, respectively, after 6 years of access to the MHVS. The sample size was insufficient to detect smaller associations with adequate precision. Additionally, we cannot rule out the possibility of measurement error, although it was likely nondifferential by treatment group, or unmeasured confounding by concomitant interventions that were implemented differentially in treated and control areas.

## Conclusions

In this study, we found that the introduction of the MHVS was positively associated with the probability of delivering in a health facility, but despite a longer period of follow-up than most extant evaluations, we did not observe attendant reductions in stillbirth, neonatal mortality, or infant mortality. Further work and engagement with stakeholders is needed to assess if the MHVS has affected the quality of care and health inequalities and whether the design and eligibility of the program should be modified to improve maternal and neonatal health outcomes.

## Author summary

### Why was this study done?

- Most women in Bangladesh do not receive essential maternal health services, with especially low rates of coverage for poorer households in rural areas.

- Starting in 2006 to 2007, the Government of Bangladesh implemented the Maternal Health Voucher Scheme (MHVS) to increase demand for and equitable use of maternal health services.

- Prior studies suggest that access to the MHVS, consistent with the broader literature on maternal voucher programs, was associated with greater use of priority maternal health services.

- However, it is unclear if short-term increases in maternal health service use are sustained over time and evidence for impacts on perinatal, infant, and maternal health outcomes, including mortality, is inconclusive.

## What did the researchers do and find?

- We linked details on the rollout of the MHVS across subdistricts (i.e., upazilas) to information on pregnancies and live births reported by women surveyed as part of the Bangladesh Demographic and Health Surveys between 2000 to 2016.

- We used a difference-in-differences design to evaluate the association between upazila-level access to the MHVS and maternal health services utilization, stillbirth, and neonatal and infant mortality.

- We observed increases in the use of maternal health services, particularly on the probability of delivering in a health facility, that materialized 2 or more years after program implementation.

- However, improvements in stillbirths, neonatal, and infant mortality were not demonstrated, raising important questions about program implementation.

## What do these findings mean?

- Potential explanations of the gap between increased services and improved outcomes include that the program may not have been targeted to reach the highest risk mothers, that implementation may have failed to increase use by mothers at highest risk, that in the absence of simultaneous supply-side interventions quality of services provided may have declined with increased numbers of patients or been inadequate, and that hospital births are associated with higher rates of bottle feeding in Bangladesh and lower rates of protective breastfeeding.

- Additionally, it is possible that small but consequential effects in health outcomes could not be discerned due to imprecision in these estimates.

- Further research is needed to examine why increased services have not yet translated into improved health outcomes as it pertains to the MHVS and other voucher programs.

- This could include cluster randomized experimental designs with the potential to address the main limitations of this study, specifically the potential for observational studies to conflate the impact of a voucher program with the effects of other population-level health and social interventions.

## Introduction

Progress in population health in Bangladesh has been called a paradox. Despite high rates of poverty, low per capita gross domestic product (GDP), and modest health systems investments [1,2], the country has made substantial progress in health since its independence in 1971, including dramatic reductions in under 5 and maternal mortality [1,3]. Despite its impressive progress, rates of maternal and neonatal mortality, estimated at 173 per 100,000 live births in 2017 and 17 per 1,000 live births in 2018 [4,5], respectively, remain high in Bangladesh compared to 2030 Sustainable Development Goal (SDG) targets. Similarly, many women are not

receiving essential maternal health services, with especially low rates of coverage for poorer households in rural areas [6–8]. According to the 2017 to 2018 Demographic and Health Surveys (DHS), among women ages 15 to 49 years who reported a live birth in the preceding 3 years, only 44% received 4 or more antenatal care visits, 49% delivered in a health facility, and 53% were assisted at delivery by medically trained personnel [9]. Further improvements in maternal and newborn health necessary to achieve the SDGs in these areas will require expanded coverage of quality reproductive and maternal health interventions that reach underserved populations [10,11].

Supply-side funding has been the primary strategy for increasing access to health services in Bangladesh, with the government providing funds to public facilities to provide free or highly subsidized maternal care [12]. Nonetheless, access is not universal, due in part to the inadequate supply of skilled health professionals [13], unofficial user fees, and discrimination by public providers against poor women [14,15], with many turning to private facilities [16]. Further supply-side investment is unlikely to reach socioeconomically disadvantaged groups without accompanying efforts that address demand-side barriers [2,8,11]. These barriers include distance to facilities, as well as travel costs and other out-of-pocket expenditures (e.g., medicines) that are not covered [16,17], and can be substantial even at public facilities [18,19]. Cultural beliefs and norms, lack of awareness about sources of care, and social stigma also shape demand for maternal health services [12].

Demand-side funding schemes are designed to increase the quantity and quality of maternal health services used by underserved populations by reducing financial barriers to access and increasing competition in the market. One example is Bangladesh's Maternal Health Voucher Scheme (MHVS), which provides pregnant women with vouchers that can be exchanged for health services from eligible public and private sector providers who are later reimbursed.

The MHVS was implemented by the Bangladesh Ministry of Health and Family Welfare (MOHFW), with technical and financial support from several multilateral and bilateral organizations, and aims "to increase demand and utilization of maternal health services, to improve access to and utilization of safe delivery, to encourage institutional delivery, and to improve equity in the utilization of maternal health services," with the ultimate goal of improving maternal and neonatal health outcomes [20,21]. It was introduced in 2006 to 2007 as a pilot program in 21 upazilas, which are subdistrict administrative units with a local governance structure that direct development activities within their boundaries [22]. By 2010, the MHVS expanded to cover 53 of Bangladesh's 545 upazilas (based on 2011 census boundaries), with another 2 upazilas gaining access in 2017 to 2018. According to the program's design, the MHVS was prioritized to upazilas with greater need based on their sociodemographic profile (e.g., literacy rate, poverty level), as well as the presence of health workers to help administer the program. However, in practice, the program was not geographically targeted based on poverty rates [21]. Within upazilas selected to participate in the program, individual-level access was initially universal in some districts and means-tested in others, but gradually shifted to a means-tested program in all upazilas. To enroll beneficiaries, community health workers identified pregnant women in their first trimester and applied to a local committee to determine if eligibility criteria were met based on residence in the upazila, parity, income, and land and asset ownership (**Table 1**). These criteria, however, were not consistently enforced. For example, vouchers were sometimes given to women who were not pregnant for the first or second time and poverty criteria were difficult to verify, resulting in imperfect targeting to poorer households [21,23]. Community health workers assisted with the distribution of program information, vouchers, referrals, subsidies, and cash incentives.

**Table 1. Bangladesh MHVS eligibility criteria and benefits[1].**

| Eligibility | Benefits |
|---|---|
| • Residence in program (treated) upazila | • Three antenatal care checkups |
| • Pregnancy is the first or second, with a family planning method used between pregnancies | • Delivery by skilled attendant (home or facility); beneficiaries received a Tk. 500 (USD 7.20)[2] incentive for delivery in home and Tk. 2,000 (USD 28.80) incentive for delivery in a facility |
| • Household income Tk. 3,100 (USD 44.64) or less per month | • Management of complications including cesarean section by designated providers |
| • Functionally landless (<6,534 ft$^2$ or approximately 0.15 acres of land) | • Postnatal care checkup within 6 weeks |
| • Household does not own income-earning assets (e.g., livestock, rickshaw) | • Transportation costs up to Tk. 500 (USD 7.20) |
| | • Gift box worth Tk. 500 (USD 7.20) |

[1]Information from Khan and Khan (2016) [21].

[2]Based on a 2010 annual average exchange rate of 0.0144 USD for each Tk.

MHVS, Maternal Health Voucher Scheme; Tk., Bangladesh Taka; USD, US dollar.

Specific services and fees covered by vouchers included: consultation fees for up to 3 antenatal care checkups, delivery by a skilled birth attendant at the home or a health facility, management of complications including eclampsia management and delivery by cesarean section, and a single postnatal care checkup within 6 weeks of delivery. Services were covered if received from accredited public, nongovernmental, or private sector providers with the capacity to provide emergency obstetric care. Providers were reimbursed through locally assigned banks after submitting evidence of service provision. Beneficiaries were reimbursed up to 500 Taka (Tk.) (7.20 US dollars (USD) based on a 2010 annual average exchange rate of 0.0144 USD per Tk.) for transport to antenatal care visits, delivery, and postnatal care services and to cover referral fees from subdistrict to district-level facilities. Additionally, beneficiaries received a cash payment of Tk. 2,000 (USD 28.80) for delivering in a health facility or Tk. 500 (USD 7.20) for delivering at home with a skilled attendant, as well as a baby-care gift box valued Tk. 500 (USD 7.20). Although it is primarily a demand-side funding scheme, the program also aims to address the supply-side through payments to public sector facilities to increase the availability and quality of maternal health services. Further details regarding the design and operation of the MHVS are available elsewhere [20,21].

Subnational, cross-sectional comparisons of voucher recipients versus nonrecipients, as well as mothers who had recently given birth in upazilas where the program was offered versus not, suggest that the MHVS was associated with greater use of maternal health services and more comprehensive maternal care, with inconsistent evidence on out-of-pocket expenditures [18,20,24–27]. However, this research cannot inform inference regarding the causal effect of the MHVS. The only quasi-experimental evaluation, a differences-in-differences analysis that compared changes in outcomes for 11 upazilas before and after they were added to the program in 2010 to corresponding changes from 11 matched control upazilas, suggests that the program increased use of public facilities, but did not influence the proportion of women receiving maternal health services unless the treated group was restricted to 5 high-performing upazilas [28,29].

In this study, we used national information on pregnancies and live births reported by women surveyed as part of the 2004, 2007, 2011, 2014, and 2017 to 2018 Bangladesh Demographic and Health Surveys (BDHS) to evaluate the association between upazila-level access to the MHVS and maternal health services utilization, stillbirth, and neonatal and infant

mortality. Our evaluation of the MHVS was designed to address several knowledge gaps concerning the impact of maternal voucher programs, including: (1) if short-term increases in the use of priority maternal health services are sustained over time [30,31]; and (2) whether they affect maternal and neonatal health outcomes [30–32], where population-level evidence is limited [30,33].

## Methods

### Study design

We used a difference-in-differences (DD) approach [34–36] to estimate the "intention-to-treat" (ITT) effect of providing upazila-level access to the MHVS (the treatment) on our outcomes of interest. The standard DD estimate compares outcome trends before versus after an intervention with corresponding trends among a "control" group, which represents the counterfactual. Since access to the MHVS was expanded to treated upazilas over the course of approximately 4 years (i.e., variation in treatment timing), we used event study models to: (1) assess whether outcome trends in treated and control areas were parallel in the preintervention period; and (2) estimate the effect of providing access to the MHVS in treated upazilas (i.e., the average treatment effect on the treated) [37,38]. A prospective analysis plan, available in **S1 Protocol**, was used in designing the study, but was not preregistered or published. In contrast to the analysis plan, we added stillbirth as an outcome and did not evaluate whether the program has affected socioeconomic inequalities in our primary outcomes, due to inadequate sample sizes and the imprecision of stratified estimates. In addition, we incorporated event study models and inverse probability weights in our DD analyses to assess evidence for parallel preintervention trends and construct more comparable treatment groups. This study is reported as per the Reporting of studies Conducted using Observational Routinely-collected Data (RECORD) guidelines (**S1 Checklist**).

### Data

Individual-level data were derived from the 2004, 2007, 2011, 2014, and 2017 to 2018 rounds of the BDHS, which are conducted in coordination with the worldwide DHS program [39]. The BDHS are repeated cross-sectional household surveys that provide information on the sociodemographic, health, and nutritional profile of the population [40]. The BDHS uses a stratified, multistage cluster sampling scheme based on the Bangladesh census and covers the entire population residing in noninstitutional dwellings. In the first stage, enumeration areas (EAs), typically city blocks in urban areas and villages in rural areas, are selected with the probability of selection proportional to EA size. In the second stage, 30 households were randomly sampled from selected EAs. Based on the household roster, all ever-married women (ages 15 to 49 in 2014 and 2017 to 2018, 12 to 49 in 2011, 10 to 49 in 2007, and 10 to 49 in 2004) who were usual members of a selected household were asked about their contraceptive use, reproductive history, and maternal health services use, among other factors. Comparability of data across waves is enhanced through interviewer training and standardized survey methods. Response rates for eligible women ranged between 97.9% and 98.8% across the 5 waves. Further details regarding the sampling, survey, and quality control procedures are available elsewhere [9,40–43]. For our analyses, we used information reported by women ages 15 to 49 to create samples of pregnancies and live births during the study period from 2000 to 2016. Details regarding sample selection are reported in **S1 Table**. The Institutional Review Board of McGill University reviewed and approved this study (IRB Study Number A05-E42-17B).

## Measures

**Treatment.** The treatment, measured at the upazila level, was gaining access to the MHVS. The list of the 55 treated upazilas that introduced the MHVS during the study period was available through public reports [20,21,29]. However, the month and year of implementation was not recorded for all treated upazilas. Through personal communication (AQ) with government officials with direct knowledge of the MHVS, we obtained the years of implementation for each upazila. We were advised to assume a starting month of December, which is the midpoint of the financial year, since it provides time for funds allocated to the program to be dispersed and the program to be implemented. Because access to the program was not randomized across upazilas, we used upazila-level administrative data (S2 Table) from the World Bank's Bangladesh Interactive Poverty Maps [44] to model the probability of treatment and create more exchangeable control groups.

We used ArcGIS software to map the location of each respondent's EA, as a proxy for the location of their residence, to its corresponding upazila based on Global Positioning System coordinates from the center of the EA, which are provided by the BDHS. This allowed us to identify if each respondent lived in a treated or control upazila and, if treated, the timing of recorded pregnancies and live births in relation to program implementation. Fig 1 shows MHVS implementation across upazilas. We added a buffer of 2.5 km around the geographic boundaries of treated upazilas, in order to account for the random displacement of EAs in publicly available DHS data and potential spillovers, as discussed in **Appendix A** in **S1 Appendix**.

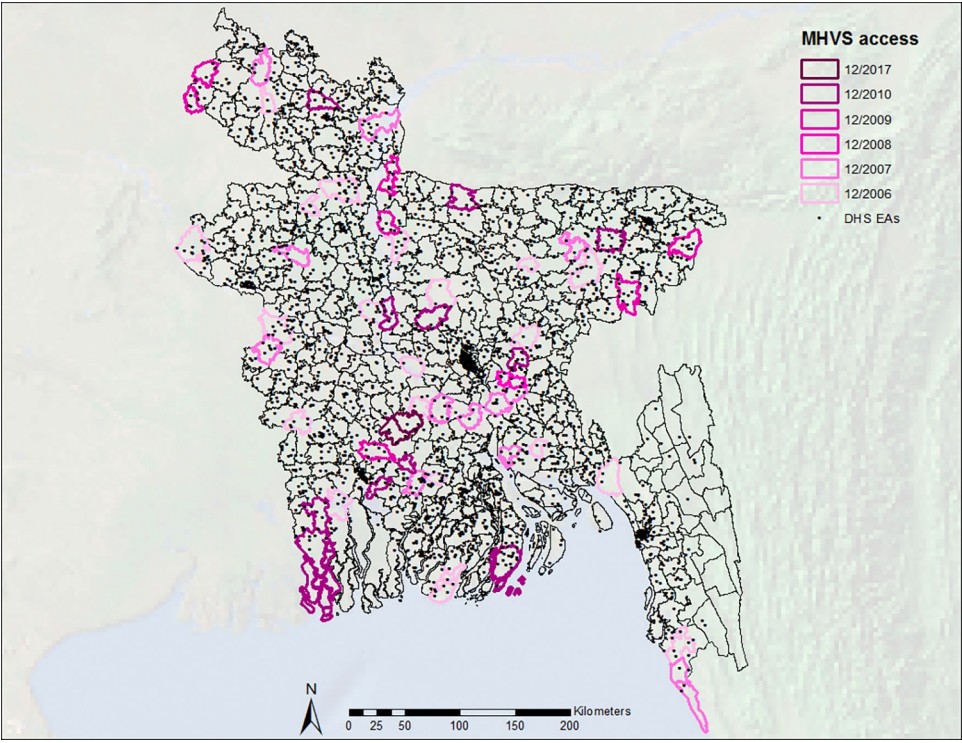

**Fig 1. Implementation of the Bangladesh MHVS by year across subdistricts (upazilas) and locations of BDHS EAs, represented by points.** Base layer extracted from the GADM database (www.gadm.org) version 3.6 (April 2018), available here: https://geodata.ucdavis.edu/gadm/gadm3.6/shp/gadm36_BGD_shp.zip. BDHS, Bangladesh Demographic and Health Survey; EA, enumeration area; MHVS, Maternal Health Voucher Scheme.

 

**Outcomes.** Indicators of maternal health services utilization, with respect to live births, included whether women reported: at least 3 antenatal care visits, which is consistent with the number of visits covered by the MHVS; having a skilled birth attendant at delivery, including a qualified doctor, nurse, midwife, paramedic, community skilled birth attendant, or family welfare visitor [45,46]; delivering in an institution or health facility outside the home, including public, nongovernmental, and private medical hospitals, clinics, and health centers; and whether the delivery was by cesarean section. We defined stillbirth as a pregnancy lasting at least 7 months that did not result in a live birth [47]. We created binary indicators for neonatal and infant mortality to record deaths occurring within the first 28 days and 1 year of life, respectively, among live births.

**Other covariates.** We included household and individual-level characteristics that might be associated with the outcomes of interest. Demographic characteristics included women's age at marriage, household size (categorized as 1 to 4, 5 to 6, or 7 or more people), rural/urban residence, and division of residence. Socioeconomic characteristics included women's and husbands' educational attainment (categorized as none, primary, or secondary or higher) and a measure of household wealth (dichotomized as above or below the median value), which is based on ownership of specific assets, environmental conditions, and housing characteristics and was constructed by DHS using the method proposed by Filmer and Pritchett [48,49]. Pregnancy and birth-related characteristics included women's age at the time of the stillbirth or live birth, measured continuously, a binary indicator for whether the interval between the index birth or pregnancy outcome and a prior birth was short (<24 months) or not (24 + months and first births), and the number of prior stillbirths.

## Statistical analyses

Event study models are an extension of the standard 2-way fixed effects DD model in which the treatment effect is estimated by a series of leads and lags representing the timing of each observation (recorded pregnancies and live births) in relation to when the MHVS was introduced in the respondent's upazila of residence:

$$Y_{ict} = \alpha_0 + \sum_{l=\leq-6}^{-2} \mu_l A_{ct}^l + \sum_{l=0}^{\geq6} \mu_l A_{ct}^l + \tau_p + \omega_t + \sum_{d=1}^{6} \gamma_d D_d + \beta_j' X_{ict} + \varepsilon_{ict}. \tag{1}$$

In the linear probability model above, we estimated for observation $i$ in upazila $c$ in year $t$ the effect of upazila-level access to the MHVS on the probability of each outcome, $Y_{ict}$, where the treatment, $A_{ct}^l$, is modeled as a series of 2-year event periods; the indicators for these periods, $l$, included 4 leads (i.e., 6+, 4 to 6, 2 to 4, and 0 to 2 years prior to the introduction of the MHVS) and 4 lags (i.e., 0 to 2, 2 to 4, 4 to 6, and 6+ years after the introduction of the MHVS), with the 2-year period prior to implementation taken as the reference. Observations from upazilas that did not receive access to the MHVS during the study period were assigned zeros for each indicator.

To account for unmeasured time-fixed differences between treatment groups, we included a fixed effect, $\tau_p$, that indicated whether the observation came from an upazila that gained access to the MHVS during the study period or not. Because the BDHS resamples upazilas in each round, we could not include upazila-level fixed effects. However, as described in **Appendix B** in **S1 Appendix**, we estimated the probability that an upazila gained access to the MHVS program during the study period, conditional on upazila-level sociodemographic characteristics, restricted the sample to the region of common support, and weighted models by the stabilized inverse probability of treatment weight [50,51]. We included fixed effects for 6 divisions (based on the 6 divisions represented in the 2004 and 2007 surveys), $\gamma_d$, to account for inter-

division differences that might be correlated with the treatment and outcomes of interest. We also included fixed effects for year of birth (or the pregnancy outcome in analyses of stillbirth), $\omega_t$, to account for secular trends in outcomes during the study period shared between treated and control upazilas. We controlled for a vector of time-varying individual-level covariates, $X_{ict}$, taken at the time of survey and assigned to each observation in year $t$, including socioeconomic, pregnancy, and birth-related characteristics. Conditional on the other covariates in the model, the coefficient for each event period, $\mu_l$, represents the difference in the probability of the outcome between treated and control observations in the event period relative to that difference in the reference period. Model (1) was used to examine potential violations of the parallel trends assumption [37,38]. In addition, our study design assumes no unmeasured time-varying confounding and no residual confounding, after weighting, by fixed differences between upazilas within divisions [36,52].

We also estimated the overall DD effect using a standard 2-way fixed effects model that replaced the event periods in Eq 1 with a time-varying indicator, $A_{ct-0.75}$, equal to 1 if the observation was from a treated upazila after it had gained access to the MHVS, and lagged by 9 months since observations measured directly after the upazila gained access to the MHVS would not have been eligible for the program.

To assess the robustness of our main findings, we compared our main event study estimates with a 2.5 km buffer around each treated upazila to those with no buffer and a larger 5 km buffer. Additionally, we compared our main results from weighted event study models to unweighted estimates. All models incorporated cluster robust standard errors to account for clustering of observations within upazilas [53].

## Results

Among women 15 to 49 years of age at the time of their interview, we restricted our samples to pregnancies and live births occurring in the 5 years prior to interview during the study period from 2000 to 2016; sample sizes ranged from 24,699 live births for analyses of antenatal care to 37,909 pregnancies lasting at least 7 months for analyses of stillbirth (**S1 Table**), from women in 508 upazilas (52 treated and 456 control). After excluding observations with missing outcome or covariate information (ranging from 2.8% to 3.3% across outcomes), sample sizes for complete case analyses ranged from 23,947 live births for analyses of antenatal care to 36,677 pregnancies for analyses of stillbirth.

Sample characteristics for live births and pregnancies prior to December 2006, when the MHVS was first introduced in treated upazilas, are presented in **Tables 2** and **3**, respectively. In general, women from areas that gained access to the MHVS were more socioeconomically disadvantaged, in terms of educational attainment and household wealth, and were more likely to come from rural areas. Women from treated areas were also less likely to report use of health services than women in control areas, including at least 3 antenatal care visits (22.2% versus 32.8%), delivering in an institution or health facility (7.2% versus 15.3%), and having a skilled birth attendant at delivery (10.8% versus 19.2%). Treated and control groups were similar in their distributions of neonatal mortality (4.0% in both groups), infant mortality (5.8% among treated versus 6.4% among control), and stillbirth (3.5% among treated versus 3.2% among control). Trends in maternal health services utilization, stillbirth, and neonatal and infant mortality before, during, and after MHVS implementation are shown in **Fig 2**.

After excluding upazilas outside the region of common support (**Appendix B** in **S1 Appendix**), weighted analyses included 338 upazilas (50 treated, 288 control) that were balanced in terms of measured upazila-level sociodemographic characteristics (the average standardized mean difference was reduced from 46.2% in the unweighted sample to 3.2% in the weighted

**Table 2. Sample characteristics for analyses of live births from the period prior to the introduction of the MHVS in treated upazilas (before December 2006); BDHS data.**

| | Total | | | Control | | | Treated | | |
|---|---|---|---|---|---|---|---|---|---|
| | N | Mean | SD | n | Mean | SD | n | Mean | SD |
| Household size | 11,315 | 6.5 | 3.2 | 9,698 | 6.5 | 3.2 | 1617 | 6.5 | 3.1 |
| Maternal age at marriage | 11,315 | 15.8 | 2.8 | 9,698 | 15.9 | 2.8 | 1617 | 15.7 | 2.6 |
| Respondent's age at event[1] | 11,315 | 23.8 | 6.1 | 9,698 | 23.8 | 6.1 | 1617 | 23.9 | 6.2 |
| | N | Frequency | % | N | Frequency | % | N | Frequency | % |
| Rural residence | 11,315 | 7,746 | 68.5 | 9,698 | 6,410 | 66.1 | 1,617 | 1,336 | 82.6 |
| Mother's education | | | | | | | | | |
| No schooling | 11,315 | 3,649 | 32.2 | 9,698 | 3,061 | 31.6 | 1,617 | 588 | 36.4 |
| Primary | 11,315 | 3,577 | 31.6 | 9,698 | 3,034 | 31.3 | 1,617 | 543 | 33.6 |
| Secondary or more | 11,315 | 4,087 | 36.1 | 9,698 | 3,601 | 37.1 | 1,617 | 486 | 30.1 |
| Father's education | | | | | | | | | |
| No schooling | 11,315 | 4,131 | 36.5 | 9,698 | 3,470 | 35.8 | 1,617 | 661 | 40.9 |
| Primary | 11,315 | 3,159 | 27.9 | 9,698 | 2,665 | 27.5 | 1,617 | 494 | 30.6 |
| Secondary or more | 11,315 | 4,016 | 35.5 | 9,698 | 3,555 | 36.7 | 1,617 | 461 | 28.5 |
| Household wealth >median | 11,315 | 5,351 | 47.3 | 9,698 | 4,797 | 49.5 | 1,617 | 554 | 34.3 |
| Birth interval <24 months[2] | 11,315 | 1,278 | 11.3 | 9,698 | 1,082 | 11.2 | 1,617 | 196 | 12.1 |
| 3+ antenatal care visits | 8,978 | 2,809 | 31.3 | 7,714 | 2,528 | 32.8 | 1,264 | 281 | 22.2 |
| Institutional delivery | 11,283 | 1,596 | 14.1 | 9,671 | 1,480 | 15.3 | 1,612 | 116 | 7.2 |
| Skilled birth attendant | 11,306 | 2,032 | 18.0 | 9,690 | 1,857 | 19.2 | 1,616 | 175 | 10.8 |
| Neonatal mortality | 11,248 | 448 | 4.0 | 9,639 | 384 | 4.0 | 1,609 | 64 | 4.0 |
| Infant mortality | 9,481 | 595 | 6.3 | 8,125 | 517 | 6.4 | 1,356 | 78 | 5.8 |

[1]Women's age at the time of the live birth, measured continuously.

[2]Binary indicator for whether the interval between the index birth and a prior birth was short (<24 months) or not (24+ months and first births).

BDHS, Bangladesh Demographic and Health Survey; MHVS, Maternal Health Voucher Scheme; SD, standard deviation.

sample, as shown in **S3 Table**). Weighted DD estimates from event study models are shown in **Fig 3**, with corresponding estimates in **S4 Table**. Each point represents the difference in the probability of the outcome comparing treated and control observations in the event period (e.g., 2 to 4 years after implementation) relative to the reference period immediately prior to implementation, along with robust 95% confidence intervals (CI).

Preintervention trends were approximately parallel for all outcomes except delivery by cesarean section, which are shown in **Appendix C** in **S1 Appendix**. In the postintervention period, there was evidence of increases in the use of maternal health services that materialized several years after the introduction of the MHVS, suggesting a lagged association. For example, after 6 years of access to the MHVS, the probabilities of delivering in a health facility and having a skilled birth attendant present increased by 6.5 (95% CI = −0.6, 13.6) and 5.8 (95% CI = −1.8, 13.3) percentage points, respectively, in treated areas compared to the control group. There was less evidence for a sustained increase in the receipt of antenatal care; 6 years after program implementation, the probability of reporting at least 3 antenatal care visits increased by 3.0 (95% CI = −4.8, 10.7) percentage points. We did not observe evidence consistent with the program having an impact on the probabilities of stillbirth, neonatal mortality, or infant mortality, with probabilities of stillbirth, neonatal mortality, and infant mortality decreasing by 0.7 (95% CI = −1.3, 2.6), 0.8 (95% CI = −1.7, 3.4), and 1.3 (95% CI = −2.5, 5.1) percentage points, respectively, after 6 years of access to the MHVS. Unweighted estimates were

**Table 3. Sample characteristics for analyses of pregnancies lasting at least 7 months from the period prior to the introduction of the MHVS in treated upazilas (before December 2006); BDHS data.**

| | Total | | | Control | | | Treated | | |
|---|---|---|---|---|---|---|---|---|---|
| | N | Mean | SD | n | Mean | SD | n | Mean | SD |
| Household size | 11,977 | 6.5 | 3.2 | 10,264 | 6.5 | 3.2 | 1,713 | 6.4 | 3.1 |
| Maternal age at marriage | 11,977 | 15.8 | 2.8 | 10,264 | 15.9 | 2.8 | 1,713 | 15.7 | 2.6 |
| Respondent's age at event[1] | 11,977 | 23.7 | 6.1 | 10,264 | 23.7 | 6.1 | 1,713 | 23.8 | 6.1 |
| | N | Frequency | % | N | Frequency | % | N | Frequency | % |
| Rural residence | 11,977 | 8,192 | 68.4 | 10,264 | 6,774 | 66.0 | 1,713 | 1,418 | 82.8 |
| Mother's education | | | | | | | | | |
| No schooling | 11,977 | 3,815 | 31.9 | 10,264 | 3,190 | 31.1 | 1,713 | 625 | 36.5 |
| Primary | 11,977 | 3,778 | 31.5 | 10,264 | 3,209 | 31.3 | 1,713 | 569 | 33.2 |
| Secondary or more | 11,977 | 4,382 | 36.6 | 10,264 | 3,863 | 37.6 | 1,713 | 519 | 30.3 |
| Father's education | | | | | | | | | |
| No schooling | 11,977 | 4,363 | 36.4 | 10,264 | 3,658 | 35.6 | 1,713 | 705 | 41.2 |
| Primary | 11,977 | 3,330 | 27.8 | 10,264 | 2,812 | 27.4 | 1,713 | 518 | 30.2 |
| Secondary or more | 11,977 | 4,273 | 35.7 | 10,264 | 3,784 | 36.9 | 1,713 | 489 | 28.5 |
| Household wealth >median | 11,977 | 5,916 | 49.4 | 10,264 | 5,283 | 51.5 | 1,713 | 633 | 37.0 |
| Birth interval <24 months[2] | 11,977 | 1,334 | 11.1 | 10,264 | 1,133 | 11.0 | 1,713 | 201 | 11.7 |
| Prior stillbirth | 11,977 | 250 | 2.1 | 10,264 | 220 | 2.1 | 1,713 | 30 | 1.8 |
| Stillbirth | 11,977 | 386 | 3.2 | 10,264 | 326 | 3.2 | 1,713 | 60 | 3.5 |

[1]Women's age at the time of the stillbirth or live birth, measured continuously.

[2]Binary indicator for whether the interval between the index pregnancy outcome and a prior birth was short (<24 months) or not (24+ months and first births).

BDHS, Bangladesh Demographic and Health Survey; MHVS, Maternal Health Voucher Scheme; SD, standard deviation.

qualitatively similar (S1 Fig), as were weighted estimates with different buffers (S2 and S3 Figs) and standard DD estimates (Fig 4).

## Discussion

Bangladesh's MHVS, introduced in 2007 and subsequently expanded to roughly 10% of the nation's subdistricts, is one of many health care voucher programs that have proliferated across low- and middle-income countries (LMICs) over the past 15 years [30]. Our analyses support 2 main conclusions. First, the introduction of the MHVS was positively associated with selected maternal health services, particularly the probability of delivering in a health facility, which is consistent with the literature on the impact of voucher schemes in LMICs [33], including Bangladesh [20,23,24,27,28]. Second, despite having a longer period of follow-up than most extant evaluations [31,33], we did not observe attendant reductions in stillbirth, neonatal mortality, or infant mortality.

Our results cohere with the few empirical studies that have examined impacts of voucher programs on measures of utilization and health outcomes [33], with prior research indicating that access to the MHVS was associated with greater use of maternal health services, including antenatal care, having a skilled attendant at delivery, delivering in a health facility, receiving postnatal care, and more comprehensive maternal care [20,24,25,27]. One cross-sectional study comparing voucher recipients to nonrecipients suggested that recipients were more likely to seek medical assistance in case of obstetric complications [54]. However, as with the broader literature on maternal voucher programs, the evidence for impacts of voucher programs on perinatal, infant, and maternal health outcomes, including mortality, is limited [30,33].

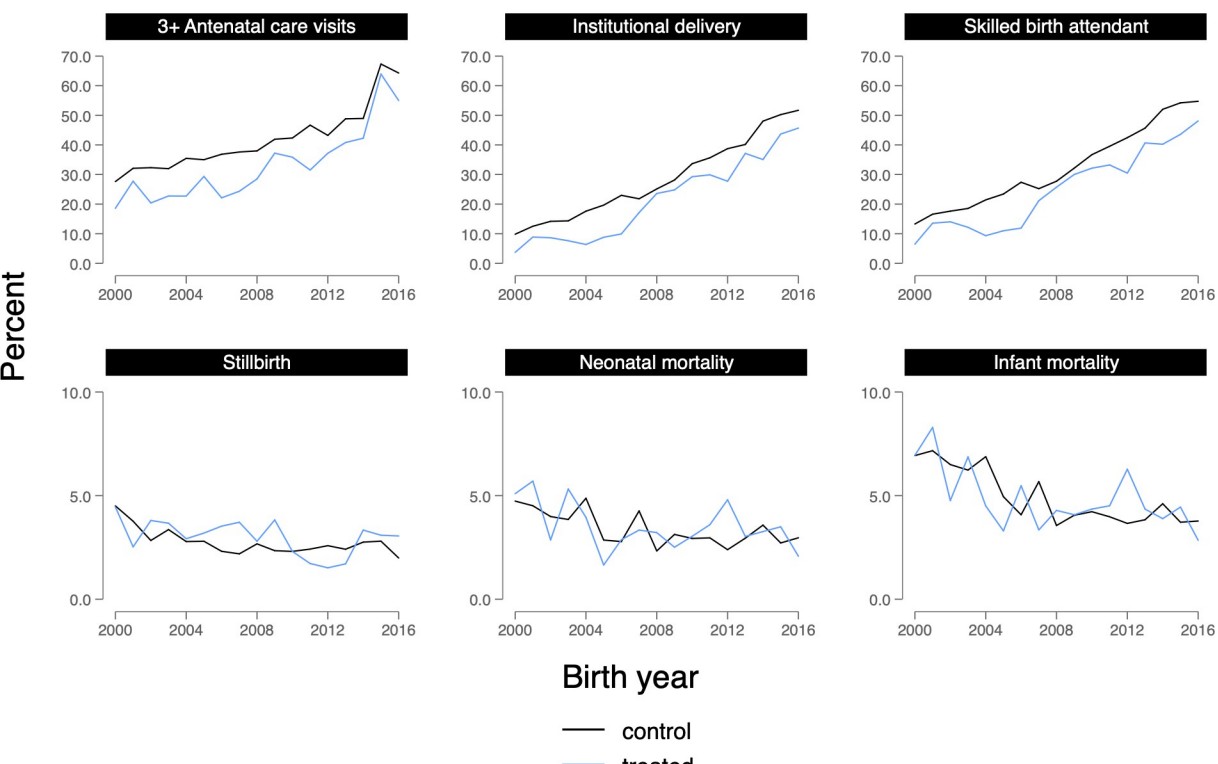

**Fig 2. Trends in measures of maternal health services utilization, stillbirth, and neonatal and infant mortality among Bangladesh MHVS respondents, 2000–2016.** MHVS, Maternal Health Voucher Scheme.

One potential explanation for why robust increases in institutional delivery and modest increases in antenatal care and skilled delivery have not been clearly associated with improved health outcomes relates to the targeting and implementation of the MHVS. Although awareness of the MHVS was high in some studies [20], it was lower in others [21,29], which may have reduced demand and utilization of the program. National reports suggest that the program distributed vouchers to 72% of the estimated number of potential beneficiaries between 2011 and 2016 [21]. However, although eligibility criteria related to pregnancy and household income were not strictly enforced [20,21,23], by design, the program may have excluded women at greater risk of adverse pregnancy and birth outcomes, including those lacking access to family planning [55].

Second, increased utilization of maternal health services may be insufficient to bring about improvements in health indicators if the quality of care remains low [56]. At least initially, the program failed to induce nonpublic providers to enroll in the program [20,23]; this may have limited competition between providers, which is a key mechanism through which demand-side funding schemes are theorized to reduce costs and increase quality. Additionally, reports indicate that providers have not been adequately compensated for services rendered or increased workload, which may have reduced the quality of available services [12,20,21,23]. Among enrolled clinics, shortages of facilities, services (i.e., emergency obstetric care), staff, and specialists were recurring issues [21,23]. These challenges, while not unique to the MHVS in Bangladesh [57], may have limited the potential for the program to have its intended effects.

Third, there may be behavioral and sociocultural explanations that warrant further investigation. For example, recent work suggests that women who delivered in health facilities were

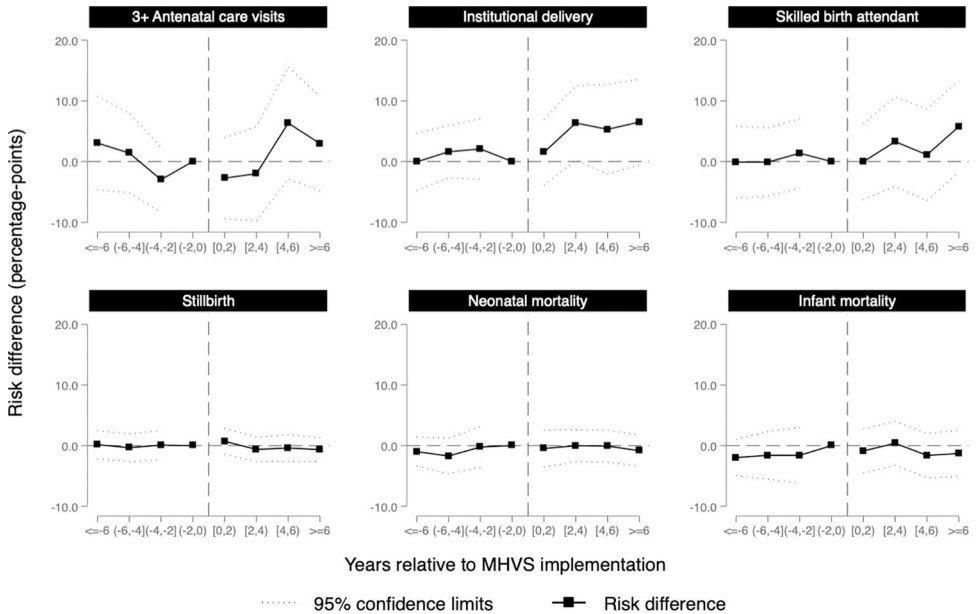

**Fig 3. Weighted event study estimates of the effect of gaining access to the MHVS in specified periods before and after the implementation of the program; BDHS data, 2000–2016.** *Notes*: Models included fixed effects for 6 divisions, fixed effects for year of birth (or the pregnancy outcome in analyses of stillbirth), and a vector of time-varying individual-level covariates, including household size, maternal age at marriage, women's age at the time of the stillbirth or live birth, rural residence, women's and husband's educational attainment, household wealth (above or below median), whether the interval between the index pregnancy or birth outcome and a prior birth was short (<24 months) or not (24+ months and first births), and prior stillbirth (in analyses of stillbirth). Weights accounted for the following upazila-level characteristics described in **S2 Table**: age structure, rural population, literacy, educational attainment, school attendance, poverty, employment, and household characteristics. Event periods included 4 leads (i.e., 6+, 4–6, 2–4, and 0–2 years prior to the introduction of the MHVS) and 4 lags (i.e., 0–2, 2–4, 4–6, and 6+ years after the introduction of the MHVS), with the 2-year period prior to implementation taken as the reference. Each coefficient represents the difference in the probability of the outcome (percentage point differences) comparing treated and control observations in the event period relative to the reference period; 95% confidence intervals account for the clustering of observations within upazilas. Sample sizes are reported in **S1 Table**, and the estimation of weights is described in **Appendix B** in **S1 Appendix**. BDHS, Bangladesh Demographic and Health Survey; MHVS, Maternal Health Voucher Scheme.

less likely than those who delivered in their homes to initiate breastfeeding within 1 hour of delivery [58], which may counteract positive effects of increased maternal health services use on infant survival through other mechanisms.

Finally, given our sample size, we were unable to detect small associations with adequate precision. An observational study comparing deliveries in 8 health facilities from treated upazilas with 8 facilities in matched control upazilas found that the incidence of stillbirth was 0.9 percentage points lower in treated facilities and concluded that "the incidence of stillbirths was significantly lower in voucher than in control areas" [20]. We found a similar effect size 6 years postimplementation, although with less precision, since our empirical approach trades off some precision in order to limit confounding [59]. Secondary data analyses that leverage existing demographic data can be viewed as a cost-effective way of generating evidence that can be triangulated with existing research.

There were other limitations to our study. With respect to measurement, the random geographic displacement of EAs may have led to nondifferential misclassification of the treatment, as described in **Appendix A** in **S1 Appendix**; in our main analyses, we added a buffer around treated upazilas that would reduce the misclassification of treated upazilas as control units and mitigate potential spillover effects, and tested different buffers from 0 to 5 km in sensitivity

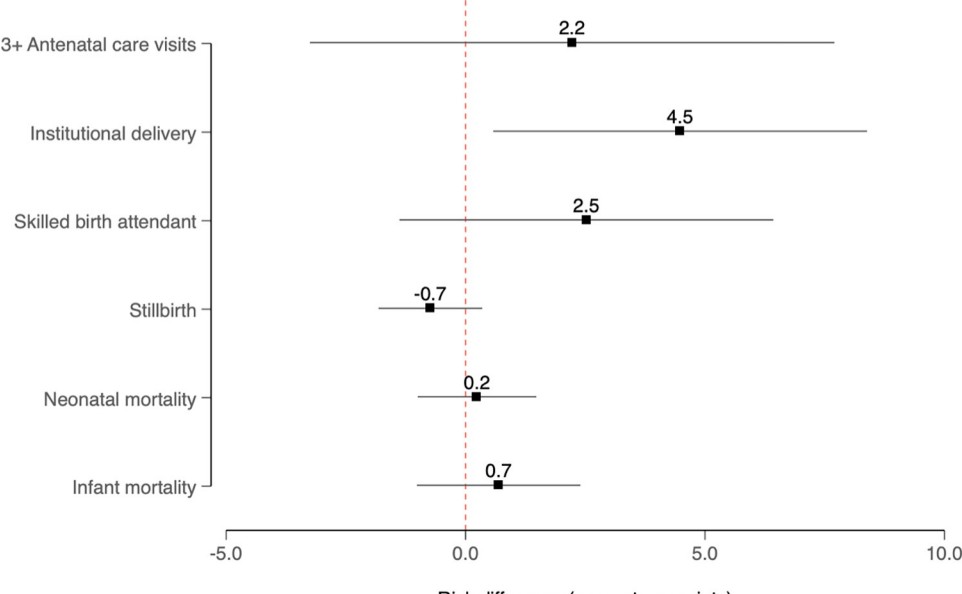

**Fig 4. DD estimates comparing changes in the treated group before and after receiving access to the MHVS relative to corresponding trends in the control group; BDHS data, 2000–2016.** *Notes*: Models included fixed effects for 6 divisions, fixed effects for year of birth (or the pregnancy outcome in analyses of stillbirth), and a vector of time-varying individual-level covariates, including household size, maternal age at marriage, women's age at the time of the stillbirth or live birth, rural residence, women's and husband's educational attainment, household wealth (above or below median), whether the interval between the index pregnancy or birth outcome and a prior birth was short (<24 months) or not (24+ months and first births), and prior stillbirth (in analyses of stillbirth). Weights accounted for the following upazila-level characteristics described in **S2 Table**: age structure, rural population, literacy, educational attainment, school attendance, poverty, employment, and household characteristics; 95% confidence intervals account for the clustering of observations within upazilas. Sample sizes are reported in **S1 Table**, and the estimation of weights is described in **Appendix B** in **S1 Appendix**. BDHS, Bangladesh Demographic and Health Survey; DD, difference-in-difference; MHVS, Maternal Health Voucher Scheme.

analyses. Additionally, the locations of respondents' EAs were measured at the time of interview, which does not account for migration between the time of each reported birth/pregnancy outcome and the time of interview. Measurement of covariates and outcomes was based on self-report, and there is likely error that is nondifferential by treatment group. For outcomes that are likely underreported, such as stillbirth, this error would tend to bias our estimates only if it were differential between our treatment and control groups. With respect to confounding bias, there is the potential for unmeasured confounding, as in any nonrandomized study. However, we included fixed effects for treatment group, year, and division and accounted for measured upazila-level characteristics using inverse probability weights to create more exchangeable treatment groups. Nonetheless, incorporating weights with event study models likely reduced the precision of our estimates and we cannot rule out the possibility of reverse confounding (e.g., if access was targeted to areas with lower levels of access to maternal health care services) or residual confounding by public, private, or nongovernmental services or initiatives that coincided with the rollout of the MHVS, differentially in treated and control upazilas. If, for example, the existing supply of community health workers in each area influenced the process by which the MHVS was phased-in across upazilas and magnified its impact on health care services use, then the effect of providing access to the program may have been over-estimated. Finally, regarding the external validity of our estimates, we selected our analytical sample based on the eligibility criteria of the MHVS, with further restriction in our main analyses to the area of common support based on propensity score distributions. We did not apply

DHS survey sampling weights and any inference beyond our sample should be done cautiously [60].

The aim of this quasi-experimental study was to provide a more comprehensive, longer-term, population-level analysis of the MHVS. Caveats considered, our results suggest that the MHVS was associated with increased utilization of maternal health services, particularly delivery in a health facility, which was incentivized by the program. However, we did not detect evidence of improvements in birth, neonatal, and infant health outcomes. Potential explanations include the suboptimal implementation of the program and inadequate supply-side investments to match increased demand and ensure access to quality health services. These challenges should be investigated and addressed [21] if the MHVS is to accelerate Bangladesh's progress toward achieving its maternal and newborn health goals.

## Supporting information

**S1 Checklist. RECORD statement: Checklist of items, extended from the STROBE statement, that should be reported in observational studies using routinely collected health data.**
(DOCX)

**S1 Protocol. Prospective analysis plan.**
(PDF)

**S1 Appendix. Appendix A, Appendix B, and Appendix C.**
(DOCX)

**S1 Table. Sample sizes for primary study outcomes.**
(DOCX)

**S2 Table. Upazila-level characteristics from the World Bank's Bangladesh Interactive Poverty Maps.**
(DOCX)

**S3 Table. Standardized mean differences comparing upazila-level sociodemographic factors in the unweighted and stabilized inverse probability of treatment weighted samples.**
(DOCX)

**S4 Table. Difference-in-differences estimates from events study models.**
(DOCX)

**S1 Fig. Unweighted event study estimates of the effect of gaining access to the Maternal Health Voucher Scheme in specified periods before and after the implementation of the program; Bangladesh Demographic and Health Survey Data, 2000–2016.**
(DOCX)

**S2 Fig. Weighted event study estimates of the effect of gaining access to the Maternal Health Voucher Scheme in specified periods before and after the implementation of the program; sensitivity analyses with 0-kilometer buffer around treated upazilas; Bangladesh Demographic and Health Survey Data, 2000–2016.**
(DOCX)

**S3 Fig. Weighted event study estimates of the effect of gaining access to the Maternal Health Voucher Scheme in specified periods before and after the implementation of the program; sensitivity analyses with 5-kilometer buffer around treated upazilas; Bangladesh**

**Demographic and Health Survey Data, 2000–2016.**
(DOCX)

## Author Contributions

**Conceptualization:** Arijit Nandi, Thomas J. Charters, Amm Quamruzzaman, Erin C. Strumpf, Jay S. Kaufman, Jody Heymann, Arnab Mukherji, Sam Harper.

**Data curation:** Arijit Nandi, Thomas J. Charters, Amm Quamruzzaman.

**Formal analysis:** Arijit Nandi, Thomas J. Charters, Sam Harper.

**Funding acquisition:** Arijit Nandi.

**Investigation:** Arijit Nandi, Thomas J. Charters, Amm Quamruzzaman, Erin C. Strumpf, Jay S. Kaufman, Jody Heymann, Arnab Mukherji, Sam Harper.

**Methodology:** Arijit Nandi, Thomas J. Charters, Amm Quamruzzaman, Erin C. Strumpf, Jay S. Kaufman, Jody Heymann, Arnab Mukherji, Sam Harper.

**Project administration:** Arijit Nandi.

**Resources:** Arijit Nandi.

**Supervision:** Arijit Nandi.

**Validation:** Arijit Nandi.

**Writing – original draft:** Arijit Nandi, Thomas J. Charters.

**Writing – review & editing:** Arijit Nandi, Thomas J. Charters, Amm Quamruzzaman, Erin C. Strumpf, Jay S. Kaufman, Jody Heymann, Arnab Mukherji, Sam Harper.

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
