## [Editor Report · Decision Letter 0]

4 Jun 2021

Dear Dr Nandi, 

Thank you for submitting your manuscript entitled "The impact of Bangladesh’s Maternal Health Voucher Scheme on health care services use, stillbirth, and neonatal and infant survival" for consideration by PLOS Medicine.

Your manuscript has now been evaluated by the PLOS Medicine editorial staff and I am writing to let you know that we would like to send your submission out for external peer review.

Please re-submit your manuscript within two working days, i.e. by Jun 08 2021 11:59PM.

Kind regards,

Louise Gaynor-Brook, MBBS PhD

Associate Editor

PLOS Medicine

---

## [Decision Letter · Decision Letter 1]

4 Oct 2021

Dear Dr. Nandi,

Thank you very much for submitting your manuscript "The impact of Bangladesh’s Maternal Health Voucher Scheme on health care services use, stillbirth, and neonatal and infant survival" (PMEDICINE-D-21-02429R1) for consideration at PLOS Medicine. 

Your paper was evaluated by three independent reviewers, including a statistical reviewer, and was discussed among all the editors here and with an academic editor with relevant expertise. The reviews are appended at the bottom of this email and any accompanying reviewer attachments can be seen via the link below:

[LINK]

In light of these reviews, I am afraid that we will not be able to accept the manuscript for publication in the journal in its current form, but we would like to consider a revised version that addresses the reviewers' and editors' comments. Obviously we cannot make any decision about publication until we have seen the revised manuscript and your response, and we plan to seek re-review by one or more of the reviewers. 

We expect to receive your revised manuscript by Oct 25 2021 11:59PM. Please email us (plosmedicine@plos.org) if you have any questions or concerns.

We look forward to receiving your revised manuscript. 

Sincerely,

Louise Gaynor-Brook, MBBS PhD

Associate Editor, PLOS Medicine

plosmedicine.org

General comments:

Please include line numbers in your revised manuscript; preferably not starting from 1 with each new page.

Please avoid inadvertently using causal language e.g. effect, effectiveness, impact, etc. 

Throughout the paper, please adapt reference call-outs to the following style: "... under five and maternal mortality [1,3]." (noting the absence of spaces within the square brackets).

Title: Please revise your title according to PLOS Medicine's style. "Impact of" should be used only if causality can be inferred, i.e., from an RCT. Please place the study design in the subtitle (ie, after a colon). We suggest “Health care services use, stillbirth, and neonatal and infant survival following implementation of the Maternal Health Voucher Scheme in Bangladesh: A difference-in-differences analysis of Bangladesh Demographic and Health Survey data, 2000-2013” or similar

Abstract Background: The final sentence should clearly state the study question.

Abstract Methods and Findings:

Please provide brief demographic details of the study population 

Please include the number of participants

Please report the main findings of your study, in keeping with the main outcome measures specified e.g. stillbirth, etc.

Please include the actual numbers / probabilities, in addition to the differences reported.

In the last sentence of the Abstract Methods and Findings section, please describe 2-3 of the main limitations of the study's methodology.

Abstract Conclusions:

Please begin your Abstract Conclusions with "In this study, we observed ..." or similar, to summarize the main findings from your study, without overstating your conclusions. Please emphasize what is new and address the implications of your study, being careful to avoid assertions of primacy. 

Author Summary:

In the final bullet point of ‘What Do These Findings Mean?’, please describe the main limitations of the study in non-technical language.

Introduction:

Please address causal language in paragraph beginning ‘In this study…’

Please remove subheadings within your Introduction 

Please relocate the section beginning “The MHVS was implemented…” to precede the paragraph beginning “Sub-national, cross-sectional comparisons of voucher recipients vs. non-recipients, …”, followed finally by the paragraph beginning ‘In this study…” (which should the the final paragraph of your Introduction) 

Methods:

Did your study have a prospective protocol or analysis plan? Please state this (either way) early in the Methods section. If a prospective analysis plan (from your funding proposal, IRB or other ethics committee submission, study protocol, or other planning document written before analyzing the data) was used in designing the study, please include the relevant prospectively written document with your revised manuscript as a Supporting Information file to be published alongside your study, and cite it in the Methods section. A legend for this file should be included at the end of your manuscript. If no such document exists, please make sure that the Methods section transparently describes when analyses were planned, and if/when reported analyses differed from those that were planned. Changes in the analysis-- including those made in response to peer review comments-- should be identified as such in the Methods section of the paper, with rationale. If a reported analysis was performed based on an interesting but unanticipated pattern in the data, please be clear that the analysis was data-driven.

Please provide more information regarding ethical approval of your study e.g. reference number

Please ensure that the study is reported according to the RECORD guideline, and include the completed RECORD checklist as Supporting Information. Please add the following statement, or similar, to the Methods: "This study is reported as per the REporting of studies Conducted using Observational Routinely-collected Data (RECORD) guideline (S1 Checklist)." The RECORD guideline can be found here: https://www.record-statement.org/ When completing the checklist, please use section and paragraph numbers, rather than page numbers which will likely no longer correspond to the appropriate sections after copy-editing.

Please note: Authors are responsible for keeping documentation of the nature and course of the personal communication with government officials.

Results: 

Please provide a single table showing the baseline characteristics of the study population.

Please quantify more of the results in the main text of your Results section.

Please provide the actual numbers of events for the outcomes where possible, in addition to summary statistics.

For the probabilities reported in your results, please specify the comparison group.

Discussion:

Please present and organize the Discussion as follows: a short, clear summary of the article's findings; what the study adds to existing research and where and why the results may differ from previous research; strengths and limitations of the study; implications and next steps for research, clinical practice, and/or public policy; one-paragraph conclusion.

Please avoid the use of causal language

Figures:

Where weighted estimates are provided, please indicate which factors are adjusted for / taken into consideration, in the respective figure legend.

Tables:

Please define abbreviations used in the table legend of each table (including Supplementary files)

Where weighted estimates are provided, please indicate which factors are adjusted for / taken into consideration, in the respective figure legend.

Tables 2 & 3 - please separate columns ‘Total’ and ‘Control’. Please present numerators and denominators for percentages. Please clarify whether the results presented are indeed Mean and SD for characteristics such as Rural Residence, Mother’s Education, Birth Interval <24 months, etc, as % would make more sense in these (and other) instances. 

References:

Please ensure that journal name abbreviations match those found in the National Center for Biotechnology Information (NCBI) databases, and are appropriately formatted and capitalised.

Please also see https://journals.plos.org/plosmedicine/s/submission-guidelines#loc-references for further details on reference formatting. For example, 6 authors should be cited prior to ‘et al’

Where website addresses are cited, please specify the date of access. 

Supplementary files: 

Please provide titles and legends for each individual table and figure in the Supporting Information.

Please see https://journals.plos.org/plosmedicine/s/supporting-information for our guidelines regarding file names.

Comments from the reviewers:

Reviewer #1: Alex McConnachie, Statistical Review

Nandi's paper looks at data from the Bangladesh Demographic and Health Surveys, to assess whether the introduction of the Maternal Health Voucher Scheme to some areas of the country had any impact on maternal health service use and outcomes. This review considers the use of statistics in the paper.

The analysis methods are generally very good. The IPTW approach is appropriate for this analysis, and the model used to assess the lagged impact of the MHVS seems good.

The main conclusion is that the MHVS led to increased use of health services, but not health outcomes. Whilst I agree that this appears to be the trend, it is notable that none of the effect estimates reported are statistically significant, so I would question the strength of the evidence at this stage. Should the wording be a little more cautious?

I feel these results are supportive of the continued rollout of the scheme, and of further analyses along similar lines in the future. This would be supported even more if there was data on the health economics of the scheme. Is that planned at some stage?

My only real question about the analysis comes from looking at Figure 2. I assume that we cannot read too much into these, as the MHVS is introduced at various times in different regions, and the curve for the "treated" line is a mixture of regions that have and have not yet received the scheme. However, the Institutional Delivery and Skilled Birth Attendant figures have a look of the intervention group lagging behind the control group, and catching up over time. This made me wonder whether the MHVS was delivered to regions partly on the basis of past/current health service utilization? If so, could some of the results simply represent regression to the mean? Were past levels of the outcomes of interest included as predictors in the propensity model used to derive the weights for the final analysis? Would this be an appropriate thing to do?

Tables 2 and 3 report the mean and standard deviation for various data items. This is unusual for categorical variables, as most of these are. I think most readers would feel more comfortable seeing these reported as frequencies and percentages.

Reviewer #2: This manuscript analyzes maternal service utilization and outcome data for delivery and newborn health across the implementation period of a national maternal health voucher system in Bangladesh. The authors have done an excellent job synthesizing the data and considered a range of potential scenarios in their sensitivity analysis and weighting approaches to counter potential gaps in the data. However, the most recent data included in this analysis is from 2014 while there was a DHS conducted in Bangladesh in 2018 - why were these data omitted from consideration as the MHVS appears to be an ongoing program? Also, while I appreciate that the data are limited to those from DHS surveys, there is mention of the MHVS program not being implemented with fidelity in all areas or by all community health volunteers in the description of the program. How were these potential irregularities considered in the analysis? How might individual CHW variation in program implementation potentially impact findings? Last, what other projects/programs were underway in Bangladesh through this period to either strengthen the supply side of the system or to further enhance patient demand? The analysis period is lengthy and in this time, there have been some notable shifts in care seeking and use in Bangladesh, principally for higher uptake of private services and a tremendous increase in Cesarean section rates. Could the authors please add these considerations to the Discussion in greater detail?

In addition to these considerations, there is a minor revisions needed: Please change the word "unawareness" to "lack of awareness" or similar on the bottom of page 3.

Reviewer #3: This is an important evaluation of a longstanding voucher scheme in Bangladesh to promote maternal health and care seeking as well as birth outcomes. Its findings are broadly commensurate with other past evaluations in that there is a small increase in coverage (and hence likely care seeking) among recipients but that there are no tangible benefits on mortality. 

Methodologically the authors did what was feasible in terms of identifying appropriate matched recipients and non-recipients for comparison. One key point that needs further highlighting is if the sample size was sufficient for mortality estimation?

The key limitation in the analysis is the supply side, so acknowledged by the authors themsleves. It would really help understand the findings if there were concomitant reasonable findings from the health systems for maternal and newborn health. These could be health workforce related, such as skilled birth attendants, facilities for transportation, newborn care etc. The distribution and coverage by CHWs or BRAC workers woukd be an additional variable. These data are available from other country analyses, and would help understand the quality of care issues that likely limited the benefits of this voucher scheme.

[LINK]

---

## [Decision Letter · Decision Letter 2]

11 Jan 2022

Dear Dr. Nandi,

Thank you very much for submitting your manuscript "Health care services use, stillbirth, and neonatal and infant survival following implementation of the Maternal Health Voucher Scheme in Bangladesh: A difference-in-differences analysis of Bangladesh Demographic and Health Survey data, 2000-2013" (PMEDICINE-D-21-02429R2) for consideration at PLOS Medicine. 

Your paper was re-reviewed by three reviewers, including the statistical reviewer, and discussed among all the editors here and with an academic editor with relevant expertise. The reviews are appended at the bottom of this email and any accompanying reviewer attachments can be seen via the link below:

[LINK]

I am afraid that we are not able to accept the manuscript for publication in the journal in its current form. In light of the comments of Reviewer 3, we will only be able to consider a revised version which includes analysis of data from the most recent 2018 Bangladesh Demographic and Health Survey. We plan to seek re-review by one or more of the reviewers. 

We expect to receive your revised manuscript by Feb 01 2022 11:59PM. Please email us (plosmedicine@plos.org) if you have any questions or concerns.

We look forward to receiving your revised manuscript. 

Sincerely,

Louise Gaynor-Brook, MBBS PhD

PLOS Medicine

plosmedicine.org

In light of the comments of Reviewer 3, please include analysis of data from the most recent 2018 Bangladesh Demographic and Health Survey in your revision, which will we require in order to reconsider your manuscript for publication.

Comments from the reviewers:

Reviewer #1: Alex McConnachie, Statistical Review

I thank the authors for their responses to my original points.

I am generally happy, though I still feel that the authors are overstating the results very slightly. The conclusion of the abstract states: "we found that the introduction of the MHVS was positively associated with the use of maternal health services". Whilst it looks likes this is probably the case, the fact remains that all three confidence intervals presented in support of this statement include zero. Ultimately, under the fitted models, the data are consistent with there being no associations. All I suggest is that this should be recognised more openly in the paper.

Reviewer #2: Thank you for these additional considerations and edits, the manuscript is now acceptable for publication.

Reviewer #3: This is a revised version of a extensively reviewed study of the potential impact of a maternal health voucher scheme (MHVS) in Bangladesh on coverage and utilization of maternal health services between 2006 (the year of the introdyuction of the scheme) and 2013. The statistical methods for assessing lagged differences using a difference in differences approach are well described, even though the sample size was likely limited for a robust assessment of impact on mortality. 

The major limitation in the usefulness of this analysis for policy is lack of a longer term assesment of impact by inclusion of data from the recent most BDHS 2018 data and I found the arguments against including this latter period in the analysis, rather facile. inclusion of a longer time period of follow up could have answered key questions on program maturity and quality as well as secular trends. The policy relevance of data that are already almost a decade old is questionable.

[LINK]

---

## [Decision Letter · Decision Letter 3]

11 May 2022

Dear Dr. Nandi,

Thank you very much for re-submitting your manuscript "Health care services use, stillbirth, and neonatal and infant survival following implementation of the Maternal Health Voucher Scheme in Bangladesh: A difference-in-differences analysis of Bangladesh Demographic and Health Survey data, 2000-2016" (PMEDICINE-D-21-02429R3) for review by PLOS Medicine.

I have discussed the paper with my colleagues and the academic editor and it was also seen again by two reviewers. I am pleased to say that provided the remaining editorial and production issues are dealt with we are planning to accept the paper for publication in the journal.

[LINK]

We look forward to receiving the revised manuscript by May 13 2022 11:59PM.   

Sincerely,

Louise Gaynor-Brook, MBBS PhD

PLOS Medicine

plosmedicine.org

Requests from Editors:

Abstract Methods and Findings:

Line 69 - please revise to ‘After six years of access…’

Abstract Conclusions:

Please briefly address the specific implications of your study, substantiated by the results.

Introduction:

Line 167 - please clarify ‘ unlikely to reach socioeconomically disadvantaged groups absent efforts’

Methods:

Thank you for providing your prospective analysis plan and further details of changes to planned analyses in your cover letter. Please include these details of where reported analyses differed from those planned in the Methods section of the paper (including those made in response to peer review comments), with rationale. 

Results: 

Line 387 - please clarify ‘3.5% among treated vs. 3.2% among treated’

Figures:

Please confirm that the GADM map is compatible with CC BY licensing

Tables:

Table S4 - please define RD, UCL and LCL in the table legend. 

References:

Your reference list appears to have been duplicated; please revise. 

Comments from Reviewers:

Reviewer #1: Alex McConnachie, Statistical Review

I thank the authors once again for considering my comments, and I am happy with this latest version. I have no further comments to make.

Reviewer #3: This revised version is much improved and the inclusion of the BDHS 2017-18 data in my opinion strengthens the analysis significantly in terms of the key finding that there does NOT seem to be any impact of the Voucher Scheme on any of the key outcomes. Irrespective of any statistical nuancing, Figure 2 is very clear in terms of lack of any substantive difference between voucher scheme recipients and counterfactuals. The final version of the paper still talks about a notable lagged effect on improved facility births, which I frankly don't see. The authors may consider a more explicit recognition of this finding not being as robust as they make out. 

If this analysis is published, as I hope it might given the importance of the findings, I would strongly recommend an accompanying commentary to discuss these counterintuitive findings and their implications for policy.

[LINK]

---

## [Editor Report · Decision Letter 4]

16 May 2022

Dear Dr Nandi, 

On behalf of my colleagues and the Academic Editor, Prof. Zulfiqar Bhutta, I am pleased to inform you that we have agreed to publish your manuscript "Health care services use, stillbirth, and neonatal and infant survival following implementation of the Maternal Health Voucher Scheme in Bangladesh: A difference-in-differences analysis of Bangladesh Demographic and Health Survey data, 2000-2016" (PMEDICINE-D-21-02429R4) in PLOS Medicine.

PRESS

Sincerely, 

Louise Gaynor-Brook, MBBS PhD 

PLOS Medicine